# Exploring *Cryptococcus neoformans* CYP51 and Its Cognate Reductase as a Drug Target

**DOI:** 10.3390/jof8121256

**Published:** 2022-11-28

**Authors:** Yasmeen N. Ruma, Mikhail V. Keniya, Brian C. Monk

**Affiliations:** 1Sir John Walsh Research Institute, Faculty of Dentistry, University of Otago, Dunedin 9016, New Zealand; 2Hackensack Meridian Health Center for Discovery and Innovation, Nutley, NJ 07110, USA

**Keywords:** *Cryptococcus neoformans*, CYP51, cognate reductase, antifungal resistance, *Saccharomyces cerevisiae* expression system, screening tool

## Abstract

*Cryptococcus* remains a leading cause of invasive fungal infections in immunocompromised people. Resistance to azole drugs has imposed a further challenge to the effective treatment of such infections. In this study, the functional expression of full-length hexahistidine-tagged *Cryptococcus neoformans* CYP51 (CnCYP51-6×His), with or without its cognate hexahistidine-tagged NADPH-cytochrome P450 reductase (CnCPR-6×His), in a *Saccharomyces cerevisiae* host system has been used to characterise these enzymes. The heterologous expression of CnCYP51-6×His complemented deletion of the host CYP51 and conferred increased susceptibility to both short-tailed and long-tailed azole drugs. In addition, co-expression of CnCPR-6×His decreased susceptibility 2- to 4-fold for short-tailed but not long-tailed azoles. Type 2 binding of azoles to CnCYP51-6×His and assay of NADPH cytochrome P450 reductase activity confirmed that the heterologously expressed CnCYP51 and CnCPR are functional. The constructs have potential as screening tools and use in structure-directed antifungal discovery.

## 1. Introduction

Inhalation of the opportunistic fungal pathogen *Cryptococcus neoformans* causes cryptococcosis, predominantly in immunocompromised individuals. *Cryptococcosis* mainly affects the lungs and can also result in meningoencephalitis, the most lethal manifestation of *C. neoformans* infection and a leading cause of death in AIDS patients [1].

*C. neoformans* is widely dispersed in nature and has typically been detected in soil, decaying wood, tree hollows, pigeon droppings and bat guano, indicative of a strong association with trees, with birds and bats as secondary hosts. *C. neoformans* occurs as three varieties and a total of four serotypes. The two varieties that predominantly cause cryptococcosis in immunosuppressed humans are *Cryptococcus neoformans* var. *grubii* (serotype A, now referred to as *Cryptococcus neoformans*) and *Cryptococcus neoformans* var. *neoformans* (serotype D, now referred to as *Cryptococcus deneoformans*). These serotypes are distinguished genetically by their *URA5* sequences [2]. While the *C. neoformans* H99 accounts for 95% of meningitis cases in HIV patients [3], *Cryptococcus gattii* (serotypes B and C) infects immunocompetent people, albeit rarely, with greatest prevalence in North America and Africa [1,4].

Using data obtained from the 2014 Joint UN Programme on AIDS for patients aged above 15 years, Rajasingham et al. [5] reported that 15% of AIDS patients die each year due to cryptococcal meningitis. Moreover, it was estimated that *C. neoformans* caused 223,100 cases annually of meningoencephalitis worldwide in HIV-infected patients, leading to about 180,000 deaths. Almost 73% of these deaths were in Sub-Saharan Africa, with 19% in the Asia and Pacific region and the remainder in North and Latin America, the Middle East and Europe [5]. It is alarming that *C. grubii* can also infect immunocompetent individuals. Studies of cryptococcal meningitis in non-AIDS patients in Vietnam [6,7], China [8,9] and South Korea [10] found that >70% of these infections are caused by the single genotype Sequence Type 5 (ST5) of *C. grubii* [11]. Specific examples in immune competent individuals include disseminated cryptococcosis in a 30-year-old individual who had no predisposing immunosuppression [12] and in an elderly immunocompetent person [13].

Conventional treatment of cryptococcosis involves three phases: induction therapy with intravenous Amphotericin B (AmpB) with or without Flucytosine, consolidation with Fluconazole (FLC) or Itraconazole (ITC) and, finally, maintenance therapy with FLC for at least a year [14]. The WHO has recently recommended an induction therapy with a single high dose of liposomal AmpB plus flucytosine and FLC (WHO: Guidelines for diagnosing, preventing and managing cryptococcal disease among adults, adolescents and children living with HIV). Unfortunately, the long duration of FLC therapy increases opportunity for recurrence of cryptococcosis and subsequent treatment failure among AIDS patients [15,16]. FLC resistance has been reported in several clinical isolates of *C. neoformans* and *C. gattii* [17,18,19,20,21,22,23] and various mechanisms, including intrinsic FLC heteroresistance of *C. neoformans* strains, have been proposed to account for this phenomenon [24,25,26,27]. During long-term FLC therapy, these strains develop a form of adaptive resistance for high concentrations of FLC, which diminishes upon removal of the drug. The mechanism underlying this resistance was found to be disomy of chromosome 1, which contains the *ERG11* (which encodes CYP51) and *AFR1* genes, the latter being responsible for the efflux of FLC from the *C. neoformans* cell. Such genomic variability leads to altered expression and a phenotype enabling a changed response to antifungal drugs [28,29]. Deletion of the *AFR1* ABC transporter gene in strain H99 substantially reduced resistance to multiple azole drugs, while its deletion of the *AFR2* and the *MDR1* major facilitator protein genes had comparatively minor effects, with only *AFR1* and *AFR2* expression upregulated by exposure to FLC [30]. Point mutations resulting in Y145F [31,32], G344S [33], G470R [34] or G484S [35] substitutions in *C. neoformans* CYP51 (CnCYP51), the target of the azole drugs, have also been associated with FLC resistance in clinical isolates. Furthermore, the CnCYP51 Y145F mutation in the enzyme’s active site appears to be responsible for differential susceptibilities to the short-tailed azoles FLC and voriconazole (VCZ) compared to the long-tailed azoles ITC and Posaconazole (PCZ) [31].

Research aimed at developing more potent azole drugs that circumvent resistance is ongoing. The azole Ravuconazole, marketed in Japan since 2018, has been found to be effective against FLC-resistant isolates and multi-azole-resistant *C. neoformans* species with the CYP51 G344S mutation [33]. An analogue of the triazole Isavuconazole, NT-a9, was found to be an effective candidate for the treatment of cryptococcal infections [36]. Recently, the tetrazoles VT-1129 [4,37,38,39] and VT-1598 [40,41] were found to have potent antifungal activity against the *Cryptococcus* species in animal models. VT-1598 was effective alone and in combination with liposomal AmpB against *C. neoformans* strain H99 [40]. These results support the development of tetrazoles for use against cryptococcal infections.

The function of CYP51 enzymes requires a cognate NADPH-cytochrome P450 reductase (CPR). This reductase accepts electrons from NADPH using its flavin adenine dinucleotide (FAD)-binding domain and then uses its flavin mononucleotide (FMN)-binding domain to supply electrons to the heme of CYP51 [42]. The CYP51-CPR interaction is important for efficient electron transfer from CPR to CYP51 and requires the FMN-binding domain have close contact with the heme of CYP51 [43]. We have mimicked this process by using a *S. cerevisiae* host strain lacking a range of efflux pumps to functionally co-express codon-optimised versions of both the *CPR* and *ERG11* genes from the H99 strain of *C. neoformans.* This approach has provided a robust model to assess the properties of wild-type CYP51 and how existing azole drugs target this enzyme. It sets the stage for investigating the impact of CPR-CYP51 interactions, the screening of novel azole drugs and determining how mutations affect CYP51 structure and function.

## 2. Materials and Methods

### 2.1. Yeast Strains, Culture Media and Reagents

*Saccharomyces cerevisiae* strains and oligonucleotides used in the study are listed in Appendix A Appendix A, respectively. All the media, reagents and kits used are the same as specified by Ruma et al. [44].

### 2.2. Construction of S. cerevisiae Strains Expressing CnCYP51 and CnCPR

Hexahistidine-tagged CnCYP51 (CnCYP51-6×His) and CnCPR (CnCPR-6×His) were expressed from the *PDR5* and *PDR15* loci, respectively, of the host strain Y2494 using the same method as described previously [44]. Y2494 was engineered from strain AD2Δ [45,46,47], with a *GAL1* promoter replacing the promoter of the endogenous *ERG11*. Strains were also created by deleting the endogenous Sc*ERG11* from the CnCYP51-6×His and CnCPR-6×His co-expressing strains. A schematic diagram showing the detailed steps of the genetic manipulation is provided in the Appendix A. All the strains were confirmed by sequencing at the Genetic Analysis Services, University of Otago. The constructed strains are listed in Table 1.

### 2.3. MIC_80_ Determination Assay

Microdilution assays were carried out in SD medium buffered to pH 6.8 to determine MIC_80_ values from the growth of individual recombinant yeast strains in response to each azole drug compared to the no drug control [47], as described previously [44]. Yeast cells were seeded at OD_600nm_ = 0.01 into serially diluted azole drugs in microtiter plates, incubated at 30 °C with shaking at 200 rpm and the cell densities measured after 48 h using a Synergy 2 microplate reader.

### 2.4. Protein Expression Determination

Crude membranes were prepared using the protocol described previously [44,47]. Coomassie-stained SDS-PAGE and Western blot analysis of samples were used to detect the presence of the hexahistidine-tagged protein [44]. Mass spectrometry identification of tryptic and chymotryptic fragments using an Orbitrap-LTQ system at the University of Otago Centre for Protein Research was used to confirm the identity of the heterologously expressed proteins (Appendix A).

### 2.5. Purification of CnCYP51

Crude membranes prepared from 12–16 L cultures of *S. cerevisiae* cells grown in YPD medium were used for the large-scale purification of the recombinant proteins. Ni-NTA affinity chromatography followed by size exclusion chromatography (SEC) was used to purify CnCYP51-6×His from solubilised crude membranes using the method described by Ruma et al. [44] and Monk et al. [44,48]. The detergent n-decyl-β-D-maltoside (DM) (Anatrace) was used for solubilisation (16 mM, 10×CMC) and purification (6.4 mM, 4×CMC).

### 2.6. Spectral Characterisation and Azole Binding Assay

The absorbance spectrum of purified CnCyp51 was obtained between 250 and 600 nm. The binding of azoles VCZ and PCZ to CnCYP51 was determined from typical Type II absorbance difference spectra recorded between 350 and 500 nm, as described by Ruma et al. [44]. Data from drug saturation curves were fitted to the Hill equation using GraphPad Prism 9 (GraphPad, San Diego, CA, USA), where Δ*A*_max_ is the maximum change in the absorbance, [azole] is the azole concentration and *n* is the Hill coefficient.

Hill equation: Δ*A* = Δ*A*_max_ [azole]*^n^*/( [azole]*^n^* + *K_d_^n^*)

### 2.7. NADPH-Cytochrome P450 Reductase Assay

To confirm that co-expressed CnCPR-6×His was active, the absorbance spectra of NADPH-cytochrome c reduction activity by CPR were determined as described by Huang et al. [49] with slight modification. Preparations containing CnCPR-6×His and CnCYP51-6×His were Ni-NTA purified from the co-expressing strains. In a reaction volume of 200 μL, 166 μL of 0.3 M potassium phosphate buffer pH 7.7, 16 μL of P450 reductase test sample (~50 μg of protein), 16 μL of 0.5 mM horse heart cytochrome c (Sigma-Aldrich, St. Louis, MO, USA) dissolved in 10 mM potassium phosphate buffer pH 7.7, and 2 μL of 10 mM NADPH (Sigma-Aldrich) freshly prepared in milli-Q water, were combined and mixed thoroughly. The absorbance spectrum was recorded using 10 mm path length cuvettes between 450 and 600 nm with a Ultrospec™ 6300 pro UV/Visible spectrophotometer operated using Resolution Spectrophotometer PC software (Biochrom, Cambridge, UK). The reference cuvette contained phosphate buffer only. The control was a preparation from a strain expressing CnCYP51-6×His but not CnCPR-6×His.

## 3. Results

### 3.1. Expression Level Analysis

The yeast strains constructed in this study are listed in Table 1.

SDS-PAGE and Western blot analysis of crude membrane preparations confirmed the expression of the recombinant proteins CnCYP51-6×His and CnCPR-6×His (Figure 1). Crude membranes were prepared from strains with and without the endogenous *ERG11*. The SDS-PAGE profile (Figure 1a) and Western blots (Figure 1b) showed that the recombinant CnCYP51-6×His enzyme expressed from the *PDR5* locus (lanes 1 to 4) appeared as a fainter band at ~62 kDa compared to the control overexpressed ScCYP51-6×His in lane 5. The expression level of these his-tagged proteins was only 10% of that of the control protein ScCYP51-6×His (Figure 1b, lane 5). These proteins were expressed at about 50% of the of the Coomassie-stained endogenous ScCYP51 (Figure 1a, lane 6) of ~61 kDa which, as expected, was not detected on the membrane fraction using the anti-his tag antibody (Figure 1b, lane 6). In lanes 3 and 4, high intensity bands at ~82 kDa indicated the expression of the CnCPR-6×His from the *PDR15* locus. The expression of the cognate reductase CnCPR-6×His from the *PDR15* locus (lanes 3 and 4) was equivalent to that of the overexpressed ScCYP51-6×His protein (lane 5).

### 3.2. Susceptibility Assays

MIC_80_ values were measured for strains Cn1–4 and the control strain Y2411 in glucose-containing SD medium to quantify their susceptibility to a range of relevant antifungals (Figure 2, Table 2). The MIC_80_ values suggest that deletion of the native CYP51 versus its repression by the GAL1 promoter, measured during growth on glucose, did not make a detectable difference in the susceptibility of the CnCYP51 overexpressing strain (Cn1 vs. Cn2) or the strains co-expressing CnCYP51 and the cognate CnCPR (Cn3 vs. Cn4) to the azole drugs tested, AmpB or Micafungin (MCF).

A reduced susceptibility to the short-tailed azoles FLC and VCZ was noted with the CnCPR co-expressing strains. The FLC MIC_80_ values of strains Cn3 (0.40 μM) and Cn4 (0.42 μM) were approximately twice those for the strains Cn1 (0.17 μM) and Cn2 (0.20 μM) (Figure 2a, Table 2). For VCZ, the MIC_80_ values increased 3- to 4-fold from 1.5–2.5 nM to 7–9 nM for the strains co-expressing CnCPR (Figure 2b). However, susceptibilities to the long-tailed triazoles PCZ and ITC were unchanged when CnCYP51 and CnCPR were co-expressed. For all four strains expressing CnCPR, the MIC values for ITC were within the range of 54–66 nM and those for PCZ were 36–40 nM (Figure 2c,d and Table 2). The tetrazole VT-1161 (Oteseconazole) caused a response similar to the long-tailed triazoles, with susceptibility not being significantly modified by co-expression of CnCYP51-6×His and CnCPR-6×His (Figure 2e). The inhibitory concentrations of VT-1161 for strains Cn1–Cn4 (23–28 nM) was almost equal to that of the host strain Y2411 (27 nM) (Table 2) and were unaffected by CnCPR co-overexpression. In case of VT-1129 (Figure 2f), the mean MIC_80_ value of ~26 nM for the co-expressing strains Cn3 (MIC_80_ = 25.4 nM) and Cn4 (MIC_80_ = 26.5 nM) gave a 1.5-fold increase in resistance as compared to a mean MIC_80_ value of 17.4 nM for strains Cn1 (17.8 nM) and Cn2 (17 nM) (Table 2). The MIC_80_ values of MCF against the reference strain Y2411 and all the recombinant strains expressing *C. neoformans* CYP51 were similar (Figure 2g). In the case of AmpB, despite a reduction in susceptibility visible in agarose diffusion experiments for strains Cn3 and Cn4 compared to strains Cn1 and Cn2, respectively (data not shown), the MIC_80_ determinations revealed a less prominent but statistically significant 1.2-fold decrease in susceptibility between Cn1 and Cn3 but not Cn2 versus Cn4 (Figure 2h). Previously we reported that overexpression of *S. cerevisiae* or *C. albicans* CYP51 in the *S. cerevisiae* host system increases the ergosterol content of cells and reduces AmpB susceptibility whereas co-expression of their cognate reductase CPR reduced the ergosterol content resulting in a rise in AmpB susceptibility [47]. A linear relationship was observed between the content of ergosterol and amphotericin B susceptibility [47]. For the present constructs, the MIC_80_ determinations suggest only a minor change in ergosterol content due to heterologous expression.

### 3.3. Purification of CnCYP51-6×His

CnCYP51-6×His was purified from strain Cn1 by Ni-NTA affinity chromatography followed by SEC. The SEC elution profile monitored at 420 nm showed a minor peak (A) eluting at 10–13 mL and a prominent peak (B) eluting at 14–16 mL (Figure 3a). Selected fractions were pooled and 5 µL samples analysed by SDS-PAGE (Figure 3b). Protein of the expected size of 61.75 kDa (Figure 3b, lane 4) was detected in peak B.

### 3.4. Spectral Characteristics of CnCYP51-6×His

The absolute absorbance spectrum for the putative CnCYP51-6×His shows that the Soret (𝛾) maximum at 417 nm corresponds to the heme group characteristic of a cytochrome P450 enzyme [50]. The enzyme also showed the α, β and the δ spectral bands at 568, 538 and 364 nm, respectively, expected for a cytochrome P450 protein (Figure 4a). A spectrophotometric index of 0.36 (A_417nm_/A_280nm_) indicated that the SEC-purified fraction was likely to contain significant impurities. The ratio of (ΔA_390nm–470nm*/*_ΔA_416nm–470nm_) was 0.43, indicating that the heme iron was predominantly in the low spin state [51]. The similar intensities of α-568 nm and β-538 nm bands and absence of an absorbance peak in the 650 nm region indicated that the CnCYP51-6×His was primarily in the oxidised enzymatically active form [52]. The CnCYP51-6×His preparation gave a characteristic Soret peak wavelength shift to longer wavelengths characteristic of Type II heme binding for the triazole drugs PCZ and VCZ (Figure 4b). Both of the drugs showed a red shift in the Soret peak of CnCYP51 from 417 nm to 422–423 nm.

### 3.5. Azole Binding Properties of Recombinant CnCYP51-6×His

Both VCZ and PCZ bound to the CnCYP51-6×His preparation, generating characteristic Type II binding absorbance difference spectra for azole binding. Type II binding spectra are produced when the N-4 of the triazole ring co-ordinates as the sixth ligand to the heme iron of the CYP51. This causes a red shift in the Soret peak due to formation of a low spin enzyme complex with the azole [53]. The difference spectra and azole saturation curves with VCZ and PCZ are presented in Figure 5. The ∆A_max_ value of VCZ binding was 0.0297 with a peak at 427.5 nm and a trough shifting from 408 to 406 nm. With PCZ the ∆A_max_ = 0.0289, the peak was observed at 426 ± 0.5 nm and the trough shifted from 408.5 to 406.5 nm. Binding with both the inhibitors gave a narrow peak but a wider trough (Figure 5a,b), suggesting that most of the iron in the CnCYP51 enzyme was in a low spin state before inhibitor addition [54]. The saturation curve for the binding of the triazoles was plotted (Figure 5c,d) and fitted using the Hill equation. The data fit the Hill equation for both VCZ (R^2^ = 0.9977) and PCZ (R^2^ = 0.9982) binding.

The parameters derived using the Hill equation are tabulated in Table 3. The dissociation constant K_d_ of VCZ was 0.39 μM, almost 2-fold higher than that observed with PCZ at 0.20 μM, suggesting PCZ is the stronger inhibitor of CnCYP51. The K_d_ values indicate tight binding of both drugs to the enzyme, as the values are lower than the concentration of the enzyme (1 μM) present in the assay [55]. The Hill coefficients for VCZ and PCZ were 2.14 and 2.39, respectively, suggesting positive allosterism for the binding of both drugs.

### 3.6. Co-purification of CnCYP51-6×His and CnCPR-6×His

Recombinant proteins from the co-expressing strain Cn3 were co-purified by Ni-NTA affinity chromatography to enable mass spectrometry identification of these proteins and to assess reductase activity. As both the enzymes had a C-terminal hexahistidine tag, Ni-NTA affinity chromatography using agarose beads with 200 mM imidazole elution was used to co-purify CnCYP51-6×His and CnCPR-6×His. Coomassie-stained SDS-PAGE gels identified the expected dominant bands migrated in the gel at ~82 kDa for CnCPR-6×His and ~62kDa for CnCYP51-6×His (Figure 6). Mass spectrometry of the bands excised from the SDS gel (from lane 5) gave high levels of coverage and confirmed the protein sequences of the heterologously expressed CnCYP51 and CnCPR (Appendix A).

### 3.7. NADPH Cytochrome P450 Reductase Activity of CnCPR

The presence of functional CnCPR-6×His was detected using the Ni-NTA affinity-purified preparation from the CnCPR co-expressing strain Cn4. The absorbance at 550 nm showed high-level cytochrome c reduction by the CnCPR-6×His plus CnCYP51-6×His (bold line, Figure 7) indicating that the recombinant CnCPR enzyme is functional. In contrast, the equivalent control preparation from the Cn3 strain expressing CnCYP51-6×His but not CnCPR-6×His did not produce an absorbance peak at 550 nm (dotted line).

## 4. Discussion

The functional expression of CnCYP51 in *S. cerevisiae*, including co-expression with its cognate NADPH-cytochrome P450 reductase (CnCPR), together with validation using Western blots and mass spectrometry of tryptic and chymotryptic fragments of these recombinant proteins, has enabled novel and robust multi-level analysis of the CYP51 from the only basidiomycete fungal pathogen of humans [56]. We have assessed both the whole cell and in vitro response of heterologously expressed CnCYP51 to azole antifungals and obtained the functional enzyme for biochemical and structural analysis. Although it has not yet been possible to obtain crystals from the limited range of conditions tested thus far with CnCYP51-6×His, which were similar to those used previously to obtain crystals of full-length Cyp51s from *S. cerevisiae*, *C. albicans* and *C. glabrata* (Monk et al., 2014, Keniya et al., 2018), our characterisation of the enzyme provide a basis for further studies. For example, the Ni-NTA affinity co-purification of co-expressed CnCYP51-6×His and CnCPR-6×His may offer opportunities to measure more precisely the affinities of CYP51 substrates and azole drugs using the BOMCC assay of fungal CYP51 activity [57].

Codon-optimised C-terminal hexahistidine-tagged *C. neoformans* lanosterol 14α-demethylase (CnCYP51-6×His) and its cognate reductase (CnCPR-6×His) were functionally expressed under control of the same promoter (*PDR5*_pro_) from the *PDR5* and *PDR15* locus, respectively, in our *S. cerevisiae* host system. Absence of the expression of native *ERG11*, either due to negative control exerted via the *GAL1* promoter (strains Cn1 and Cn3) or following deletion of the native *ERG11* (strains Cn2 and Cn4), did not appear to affect the overall function and viability of the cells expressing CnCYP51, either with (strains Cn2 and Cn4) or without the co-expression of CnCPR (strains Cn1 and Cn3). Thus, *PDR5*-mediated expression CnCYP51 complemented the endogenous *ScERG11* gene. Furthermore, the phenotypes observed in susceptibility assays for a range of antifungal drugs in clinical use, as well as tetrazoles VT-1161 and VT-1129, showed both CnCYP51 and CnCPR are functional when heterologously expressed in the yeast host strain.

An important limitation of the study is that, despite using codon optimisation in designing our constructs for expression in *S. cerevisiae* under control of the gain-of-function pdr1-3 transcriptional regulator acting at the *PDR5* locus, only modest expression of CnCYP51 was obtained. The recombinant strains were at least twice as sensitive to the triazole drugs FLC, VCZ, ITC and PCZ as the control yeast host strain (strain Y2411) which expressed the endogenous ScCYP51 (Erg11). The modest expression detected might be due to a problem caused by codon usage choice resulting in misfolding bottlenecks. In addition, evolutionary distance might affect enzyme turnover due to a sub-optimal membrane microenvironment or substrate preference, i.e., a likely preference for eburicol over the lanosterol provided by the yeast expression system [58]. Despite a low level of CnCYP51 expression and the obligatory use of endogenous lanosterol as a substrate, the normal growth rates, morphology and susceptibility to the control drug MCF of the recombinant cells and the inability to detect any suppressor strains on exposure to azole drugs suggests that the CnCYP51 constructs are not under genetic stress sufficient to modify susceptibility via increased azole target expression.

Another possible limitation is that the recombinant CnCPR is much more highly expressed than CnCYP51. However, overexpression of the cognate CnCPR as a CnCPR-6×His construct resulted in significant differences in the susceptibility of the co-expressing strains to azoles without increasing CnCYP51 expression levels. Expression of CnCPR decreased susceptibility to the short-tailed azoles FLC and VCZ by 2-fold and 3- to 4-fold, respectively, but susceptibilities for ITC and PCZ were unchanged. The different response to the short- and long-chain triazoles is probably due to the binding of FLC and VCZ being confined to the active site within the ligand-binding pocket while the binding of ITC and PCZ includes additional interactions with the substrate entry channel [45,48,59]. The susceptibility responses of the potent tetrazoles VT-1161 and VT-1129 to the co-expression of CnCPR-6×His more closely matched those of the long-tailed triazoles than the short-tailed triazoles. This finding is consistent with the weaker interaction with the heme mediated by the tetrazole group in VT-1161 and VT-1129 than for the triazole group in FLC, VCZ, ITC and PCZ [60], the likely maintenance of a water-mediated interaction between their tertiary hydroxyl groups with the hydroxyl of *S. cerevisiae* CYP51 Y140 for VT-1161 and VT-1129 in common with FLC and VCZ, and the medium-length tails of VT-1161 and VT-1129 having significant hydrophobic interactions with the substrate entry channel not accessible to FLC and VCZ [61].

The function of CPR is to provide electrons to the CYP51 enzyme during its catalytic cycle, enabling formation of a highly reactive intermediate iron (IV) oxo (or ferryl) radical complex during the P450 catalytic cycle [62]. This is achieved by a rapid association and dissociation of P450:CPR complexes in order to maintain electron transfer between the enzymes [63,64]. When CnCPR is overexpressed, it is possible that the overabundance of CnCPR we observed enables enhanced production of the iron (IV) oxo radical complex and increased substrate occupancy in the CnCYP51 active site. We speculate that this situation could prevent the competitive binding of the short-tailed azoles FLC or VCZ at the active site. In contrast, the long-tailed azoles ITC and PCZ may overcome this disadvantage due to additional interactions in the substrate entry channel of the enzyme. This interpretation suggests how overexpression of a redox partner CPR in fungal cells may contribute to differential resistance to the short-tailed azoles.

FLC-resistant *C. neoformans* strains are an ongoing concern as they are frequently found among cryptococcal meningitis clinical isolates [24,26,31,32,35,65]. The model reported here offers the opportunity to explore the impact of such mutations on CnCYP51 structure, function and interaction with azole drugs including FLC. For example, structural alignment of *S. cerevisiae* CYP51 Y140 with CnCYP51 Y145 provides the most likely explanation for the CnCYP51 Y145F mutation causing differential susceptibilities to the short-tailed azoles FLC and VCZ compared to the long-tailed azoles ITC and PCZ [31]. Further phenotypic and biochemical exploration of the impact of cognate *C. neoformans* CPR enzyme expression levels may allow a better understanding of how it affects susceptibility and resistance to the short-tailed azoles. A similar effect was previously seen with CaCYP51 co-expressed with CaCPR in yeast [47]. In several insect systems, high expression levels of CPR have been related to insecticide resistance. These effects have been found in the resistance of the fruit fly *Bactrocera dorsalisor* to malathion [66], the pest Citrus aphid *Aphis citricidus* to abamectin [49], *Tetranychus cinnabarinus* to fenpropathrin [67] and in an isoprocarb-resistant strain of *Rhopalosiphum padi* [68].

Structural characterisation of the *C. neoformans* sterol 14α-demethylase is required to gain insight into its substrate specificity and how azole drugs are bound, in order to obtain the atomic detail required for structure-directed antifungal discovery. Structural studies are also needed to address, via co-crystallisation, the possible impact of cognate reductase overexpression on azole resistance. The heterologous expression system described in the present report provides opportunity for molecular-level analysis of the interaction between fungal CYP51, its ligands and its cognate CPR.

## Figures and Tables

**Figure 1 jof-08-01256-f001:**
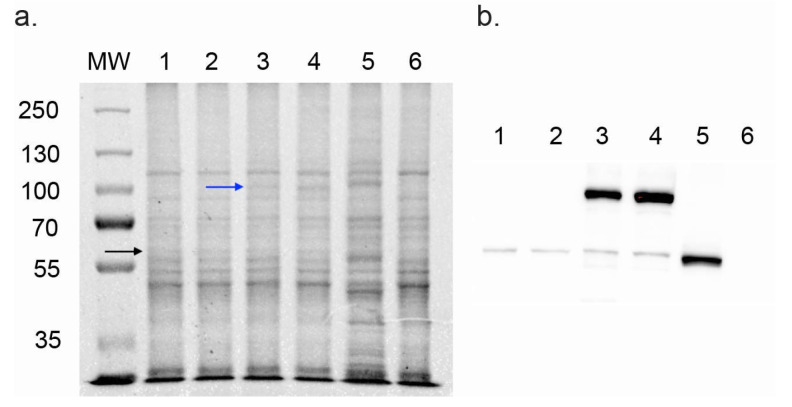
SDS-PAGE and Western blot analysis of CnCYP51-6×His and CnCPR-6×His. (**a**) Coomassie-stained SDS-PAGE gel of crude membranes (15 μg of protein) expressing recombinant *C. neoformans* proteins. (**b**) Western blots of crude membranes (15 μg/lane) decorated with mouse anti-6×His antibody and visualised using ECL. Lanes: 1. Cn1, 2. Cn2, 3. Cn3, 4. Cn4, 5. Y941, 6. Y1857 (AD2Δ). MW: 5 µL protein molecular weight markers (PageRuler Plus Prestained Protein Ladder, Thermo Scientific, Waltham, MA, USA). Molecular weights are given in kDa. Blue arrow indicates the band for CnCPR and black arrow indicates CnCYP51.

**Figure 2 jof-08-01256-f002:**
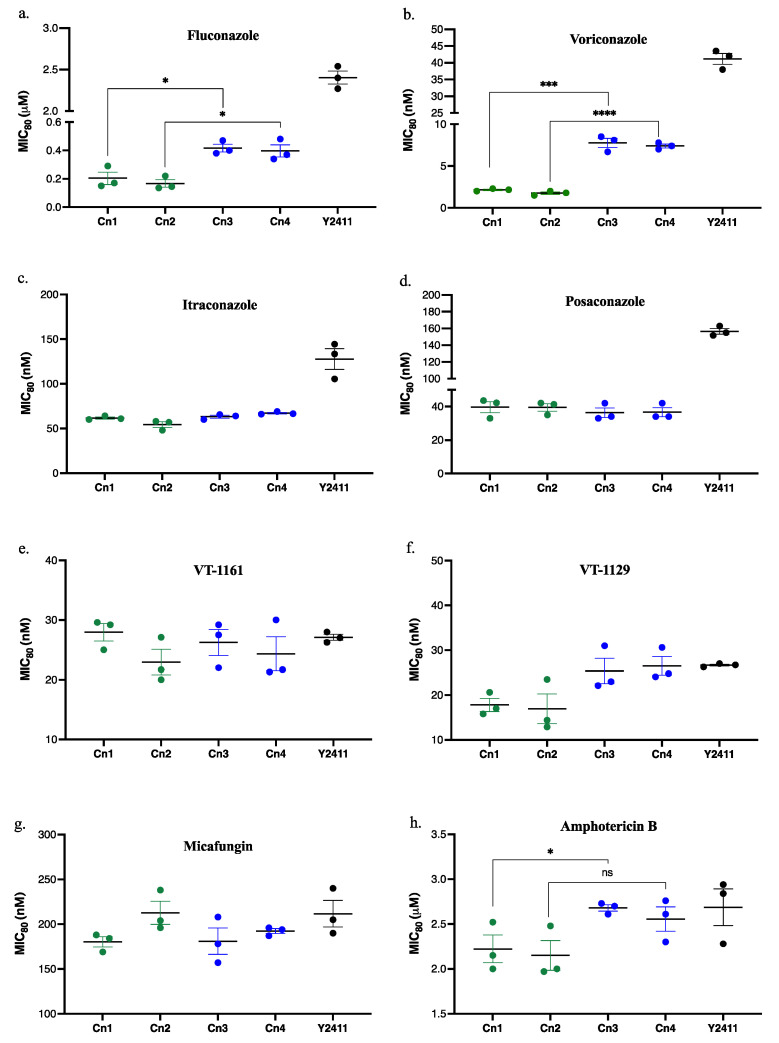
Susceptibilities of strains Cn1–Cn4 and Y2411 to azole antifungals. (**a**) FLC, (**b**) VCZ, (**c**) ITC, (**d**) PCZ, (**e**) VT-1161, (**f**) VT-1129, (**g**) MCF, (**h**) AmpB. Glucose-containing SD medium buffered at pH 6.8 was used. The MIC_80_ values were determined in microtiter plates from the OD_600nm_ values compared with no drug controls. The experiments were carried out in triplicates in three separate experiments for each strain and drug. The scatter plot is based on the average MIC_80_ values of three individual experiments (*n* = 3). The horizontal line at the centre represents the mean MIC_80_ and the error bars indicate the standard error of the mean (SEM). An unpaired two-tailed Student’s *t*-test was used to compare the MIC_80_ values of the strains, Cn1 vs. Cn3 and Cn2 vs. Cn4. The statistical significance is shown with an asterisk. * *p*  <  0.05, *** *p*  <  0.001 and **** *p*  <  0.0001. ns—non-significant. The statistical significance values for ITC, PCZ, VT-1129, VT-1161 and MCF were non-significant (not shown on plot).

**Figure 3 jof-08-01256-f003:**
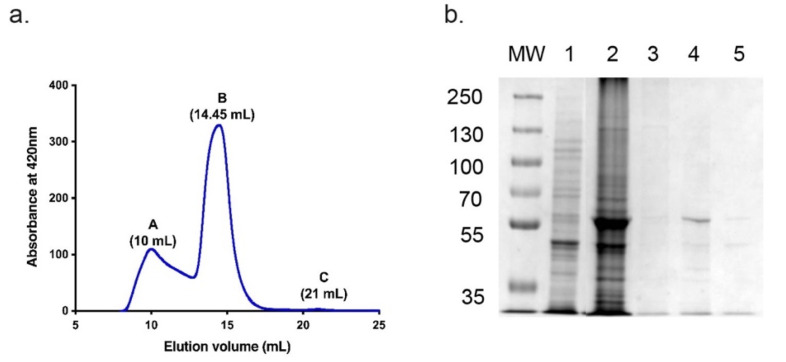
SEC fractionation of Ni-NTA affinity-purified CnCYP51-6×His. (**a**) SEC elution profile-absorbance at 420 nm. (**b**) SDS-PAGE profile of the CnCYP51-6×His purification. MW: 5 µL protein molecular weight markers (PageRuler Plus pre-stained protein ladder, Thermo Scientific). Lane 1: 20 μg of crude membrane fraction. Lane 2: 5 µL pooled Ni-NTA affinity-purified fraction. Lane 3: Peak A, 5 µL pooled 10–13 mL of the elution fraction. Lane 4: Peak B, 5 µL pooled 14–16 mL fraction. Lane 5: 5 µL pooled 17–21 mL fraction.

**Figure 4 jof-08-01256-f004:**
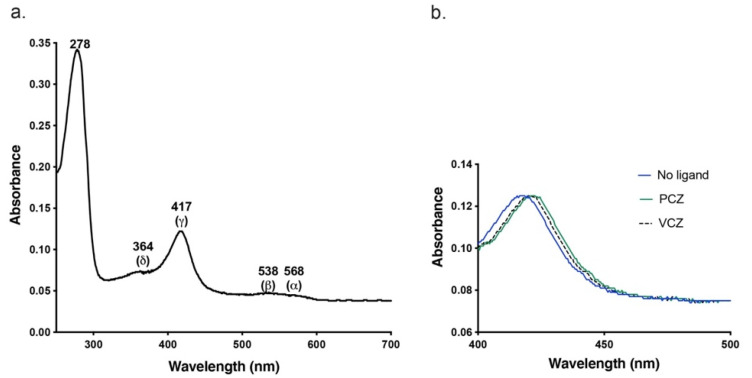
(**a**) Absolute absorbance spectrum of ligand-free SEC-purified CnCYP51-6×His. (**b**) The Soret peak of Ni-NTA affinity- and SEC-purified CnCYP51-6×His with PCZ, VCZ or no ligand.

**Figure 5 jof-08-01256-f005:**
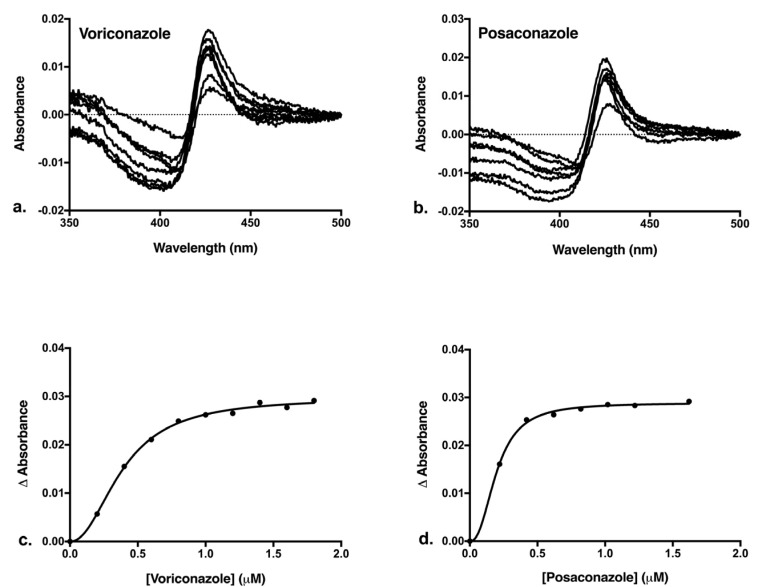
Binding of triazoles VCZ and PCZ to CnCYP51-6×His. VCZ was dissolved in Milli-Q water and PCZ was dissolved in DMSO and each titrated against 1 μM CnCYP51. (**a**) Type II difference spectra with VCZ. (**b**) Type II difference spectra with PCZ. (**c**) Saturation curve for VCZ binding fitted using the Hill equation. (**d**) Saturation curve for PCZ binding fitted using the Hill equation.

**Figure 6 jof-08-01256-f006:**
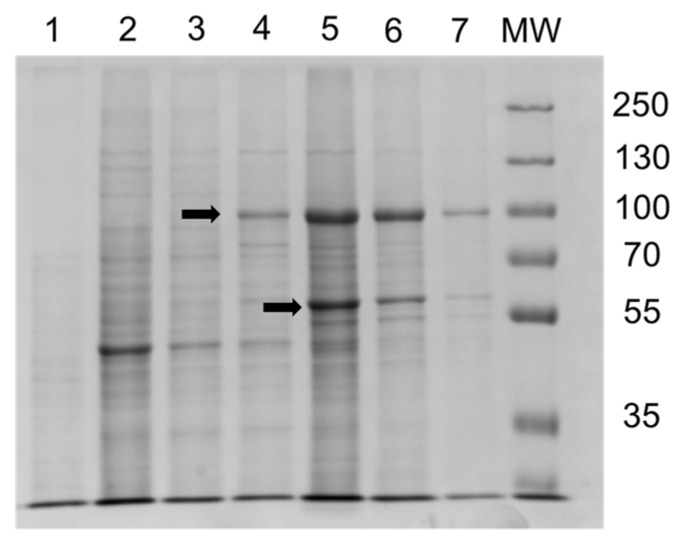
SDS-PAGE analysis of Ni-NTA agarose elution profile for CnCPR-6×His co-expressed with CnCYP51-6×His. Lane 1: 10 µL of the flow through after the DM solubilised plasma membrane fraction was incubated overnight with the agarose beads. Lanes 2 and 3: 5 µL of the 1 mM histidine washes of the proteins bound agarose beads. Lanes 4–7: 5 µL of the proteins eluted using 200 mM imidazole. MW: 5 µL protein molecular weight markers (PageRuler Plus pre-stained protein ladder, Thermo Scientific). The arrow at 90 kDa indicates the putative CnCPR-6×His (82 kDa size) and the arrow at 62 kDa indicates the putative CnCYP51-6×His.

**Figure 7 jof-08-01256-f007:**
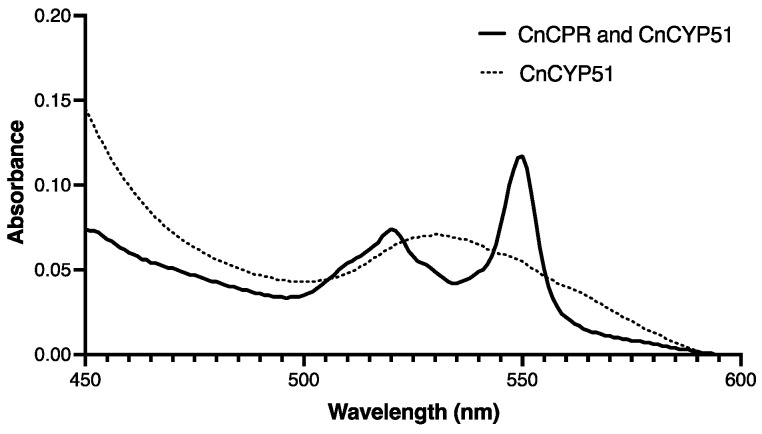
Absorbance spectra of cytochrome c reduction by CnCPR-6×His.

**Table 1 jof-08-01256-t001:** Strains constructed in the study.

Strain	Strain Number	Strain Description
Cn1	Y2708	*PDR5::*Cn*CYP51*-6×*His,* Δ*ERG11pro::GAL1pro*
Cn2	Y2709	*PDR5::*Cn*CYP51*-6×*His,* ΔSc*ERG11::HIS1*
Cn3	Y2710	*PDR5::*Cn*CYP51*-6×*His, PDR15::*Cn*CPR*-6×*His,* Δ*ERG11pro::GAL1pro*
Cn4	Y2711	*PDR5::*Cn*CYP51*-6×*His, PDR15::*Cn*CPR*-6×*His,* Δ*ScERG11::HIS1*
Cn5	Y2712	*PDR15::*Cn*CPR*-6×*His,* Δ*ERG11pro::GAL1pro*

**Table 2 jof-08-01256-t002:** MIC_80_ values for strains overexpressing CnCYP51 with/without CnCPR.

Strains	MIC_80_							
	FLC	VCZ	ITC	PCZ	VT-1161	VT-1129	MCF	AmpB
	(μΜ)	(nΜ)	(nΜ)	(nΜ)	(nΜ)	(nΜ)	(nΜ)	(μΜ)
**Y2411**	2.4 ± 0.08	41.7 ± 1.6	128 ± 12	157 ± 3.4	27.1 ± 0.5	26.7 ± 0.2	212 ± 15	2.69 ± 0.2
**Cn1**	0.2 ± 0.04	2.39 ± 0.3	62.6 ± 1.6	39.6 ± 3.3	28 ± 1.5	17.8 ± 1.4	180 ± 6	2.2 ± 0.15
**Cn2**	0.17 ± 0.03	1.8 ± 0.15	54.3 ± 3.2	39.5 ± 2.3	23 ± 2.1	17 ± 3.3	213 ± 13	2.15 ± 0.17
**Cn3**	0.42 ± 0.03	7.8 ± 0.5	65.5 ± 0.1	36 ± 2.8	26.2 ± 2.2	25.4 ± 3.0	181 ± 15	2.68 ± 0.04
**Cn4**	0.4 ± 0.04	7.4 ± 0.23	66.3 ± 0.3	37 ± 2.7	24.5 ± 3.0	26.5 ± 2.1	192 ± 3	2.56 ± 0.14

MIC_80_ values are shown as the mean values ± SEM for 3 separate clones of each strain using data obtained in triplicate measurements from at least 3 different experiments (a total of 9 measurements per strain).

**Table 3 jof-08-01256-t003:** Type II binding characteristics of triazoles to CnCYP51-6×His.

Triazole	Δ*A*_max_ ^a^	𝞴_peak_	𝞴_trough_	Hill Number ^b^	*K_d_* (μM ^b^)
Voriconazole	0.0297	427.5	408 to 406	2.14 ± 0.14	0.39 ± 0.013
Posaconazole	0.0289	425.5 to 426.5	408.5 to 406.5	2.39 ± 0.27	0.20 ± 0.01

^a^ The standard deviations of the Δ*A*max values were less than 5%; ^b^ Standard errors included for Hill number and *Kd* determinations.

## Data Availability

Not applicable.

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
