# Peer review of "Exploring Cryptococcus neoformans CYP51 and Its Cognate Reductase as a Drug Target"

_jof, 2022, doi:10.3390/jof8121256_

Round 1
Reviewer 1 Report
It is now widely accepted that C. neoformans yeasts are not only found in soils and bird droppings, this have evolved…
Please use the current nomenclature : C. neoformans var grubii was now C. neoformans and C. neoformans var neoformans was C. deneoformans.
It would be nice if all researchers working on Cryptococcus used the same nomenclature in their articles.
Line 34 « While the serotype A strain H99 accounts for 95% of meningitis cases in HIV patients »
The authors want to talk about C. neoformans, why they cited H99 strain here ?
Line 35-36 Cryptococcus neoformans var. gattii (serotypes B and C) infects immunocompetent people with greatest prevalence in North America and Africa [4].
This is not quite right
We are talking about a species complex and C. gattii, not var gattii. They are not reduced to a prevalence in Africa and North America (this was originally described, this have since evolved like the biotope)
For the first paragraph of the introduction, please refer to more recent articles.
Concerning fluconazole resistance, you could also refer to more recent articles.
About the treatment of cryptococcosis please add a reference like the recommandation of « World Health Organization» et please correct with the right treatment recommended by the WHO
CYP51 was named ERG11 in Cryptococcus
FLC resistance was due to ERG11 and AFR1 (please explain this abreviation) in C. neoformans. It’s not only AFR1 which was responsible for FLC resistance. MDR1, AFR2 ?
What is FNM ?
The aim of this publication was « We have mimicked this process by using a S. cerevisiae host strain to functionally co-express codon-optimized versions of both the CPR and CYP51 genes from the H99 strain of C. neoformans. »
What is the link with FLC resistance previously explain in the introduction
The introduction is not clear or appropriate to the purpose of the study
In table 11 : CYP and ERG, please use only ERG
The obtention of MIC80 was poorly explain in the section material and methods. Why measured the MIC80 and not the MIC50. What is the correspondance with clinical value.
The introduction should be rewritten by referencing more recent articles and using more recent nomenclature for Cryptococcus. The introduction should be rewritten in a way that is more in line with the purpose of the study. The results are clearly explained even if the figures are not very readable. Like the introduction, the discussion needs to be more precise. It hard to see the point of the study even though the methods used are complex.
Author Response
Reviewer 1
Comments and Suggestions for Authors
The reviewer is thanked for their consideration of the manuscript and useful comments.
It is now widely accepted that C. neoformans yeasts are not only found in soils and bird droppings, this have evolved…
Response: Corrected in lines 29-31
Please use the current nomenclature : C. neoformans var grubii was now C. neoformans and C. neoformans var neoformans was C. deneoformans.
Response: Corrected throughout MS
It would be nice if all researchers working on Cryptococcus used the same nomenclature in their articles.
Line 36 « While the serotype A strain H99 accounts for 95% of meningitis cases in HIV patients »
The authors want to talk about C. neoformans, why they cited H99 strain here ?
Response: H99 is the dominant and most studied strain of C. neoformans causing cryptococcosis and 95% of meningitis in HIV patients. H99 also provided the genetic information used for the constructs used in the present study
Cryptococcus neoformans var. gattii (serotypes B and C) infects immunocompetent people with greatest prevalence in North America and Africa [4].
This is not quite right
We are talking about a species complex and C. gattii, not var gattii. They are not reduced to a prevalence in Africa and North America (this was originally described, this have since evolved like the biotope)
Response: Corrected line 36-39
For the first paragraph of the introduction, please refer to more recent articles.
Response: A recent (2021) article, reference 1, cited
Concerning fluconazole resistance, you could also refer to more recent articles.
Response: 2 additional key articles cited (lines 71, 75 and 80)
About the treatment of cryptococcosis please add a reference like the recommandation of « World Health Organization» et please correct with the right treatment recommended by the WHO
Response: Most recent WHO recommendation included and referenced in lines 56-59
CYP51 was named ERG11 in Cryptococcus
FLC resistance was due to ERG11 and AFR1 (please explain this abreviation) in C. neoformans. It’s not only AFR1 which was responsible for FLC resistance. MDR1, AFR2 ?
Response:: In line 67 “CYP51” has been changed to “ERG11 (which encodes CYP51)” as requested. However, for the codon optimised (and hexahistidine tagged) for S. cerevisiae copy of C. neoformans H99 ERG11 we prefer retaining the name CnCYP51-6xHis to avoid confusion. Most papers on C. neoformans sterol 14alpha-demethylase refer to the CYP51 and not the ERG11enzyme, the rationale being that its substrate specificity has yet to be defined.
The minor contribution of AFR2 and MDR1 to FLC resistance is described and referenced (lines 69-73)
What is FNM ?
Response: FMN and FAD are defined in lines 90-92
The aim of this publication was « We have mimicked this process by using a S. cerevisiae host strain to functionally co-express codon-optimized versions of both the CPR and CYP51 genes from the H99 strain of C. neoformans. »
What is the link with FLC resistance previously explain in the introduction
Response: Two sentences added (lines 97-100) to make the link requested.
“This approach has provided a robust model to assess the properties of wild type CYP51 and how existing azole drugs target this enzyme. It sets the stage for investigating the impact of CPR-CYP51 interactions, the screening of novel azole drugs, and determining how mutations affect CYP51 structure and function.”
The introduction is not clear or appropriate to the purpose of the study
In table 11 : CYP and ERG, please use only ERG
Response: See response above
The obtention of MIC80 was poorly explain in the section material and methods. Why measured the MIC80 and not the MIC50. What is the correspondance with clinical value.
Response: As noted in the Materials and Methods section, we have extensively published MIC80 values rather than MIC50 values in our studies using heterologous expression our yeast system e.g. reference 44. This approach successfully discriminates the end point even if there is a low but significant level of background growth at fungistatic azole concentrations above the MIC80. We do not have access to clinical isolates of C. neoformans we do not consider it appropriate for us to comment on correlations with such organisms. The use of the hypersensitive S. cerevisiae host strain deleted of multiple drug efflux pumps provides a high signal to noise ratios that enable study of even relatively weakly expressed gene products of interest. In contrast, genetic variability, presence of drug efflux pumps, and their induction by azole drugs, as well as compensatory expression of homologous genes in deletion mutants, have the potential to confound studies of CYP51 in clinical isolates.
The introduction should be rewritten by referencing more recent articles and using more recent nomenclature for Cryptococcus. The introduction should be rewritten in a way that is more in line with the purpose of the study. The results are clearly explained even if the figures are not very readable. Like the introduction, the discussion needs to be more precise. It hard to see the point of the study even though the methods used are complex.
Response: The purpose of the study has been more clearly defined to align with the introduction and discussion by adding at line 97-100:
“This approach has provided a robust model to assess the properties of wild type CYP51 and how existing azole drugs target this enzyme. It sets the stage for investigating the impact of CPR-CYP51 interactions, the screening of novel azole drugs, and determining how mutations affect CYP51 structure and function.”
We believe the main point of the study – the creation and properties of a robust model system expressing a functional C. neoformans CYP51 and its cognate reductase is addressed at the relevant level of detail in the discussion. The discussion has been amended to include the potential of the system to investigate target-mediated azole (FLC) resistance in molecular detail by including at line 399:
“The model reported here offers opportunity to explore the impact of such mutations on CnCYP51 structure, function and interaction with azole drugs including FLC. For example, structural alignment of S. cerevisiae CYP51 Y140 with CnCYP51 Y145 provides the most likely explanation for the CnCYP51 Y145F mutation causing differential susceptibilities to the short-tailed azoles FLC and VCZ compared to the long-tailed azoles ITC and PCZ [31].”
Additional minor corrections highlighted
Line 2 neoformans
Line 4 Both affiliation of M V Keniya given
Line 124 Grammar in sentence corrected
Line 173 Molecular weight of endogenous ScCYP51 given
Line 204 Commercial name of VT-1161 (Otesoconazole) given
Lines 245, 253 and 254 Volumes in elution profile corrected
Line 348 Sentence simplified to summarize conclusion of previous sentence
Table 2 Compound MCC8186 deleted as not mentioned in text

Reviewer 2 Report
The authors have written the paper quite well, presenting the scientific rationale for doing the experiments in a logical manner. The stepwise approach to conducting the experiments was well described.
There are a few minor corrections, including repetitions of words (lines 141 and 160).
Need to spell out meaning of MCF in full.
Need to correct inconsistency in font (line 114).
Need to include limitations in the discussion.
Author Response
Reviewer 2
The reviewer is thanked for their consideration of the manuscript and their comments.
All comments requiring amendments were minor.
The authors have written the paper quite well, presenting the scientific rationale for doing the experiments in a logical manner. The stepwise approach to conducting the experiments was well described.
There are a few minor corrections, including repetitions of words (lines 141 and 160).
Response: Corrected in required lines
Need to spell out meaning of MCF in full.
Response: Name of MCF spelled out in line 194
Need to correct inconsistency in font (line 114).
Response: Done
Need to include limitations in the discussion.
Response: The primary limitations of the system – the modest expression of CnCYP51-6xHis and the relative overexpression of CnCPR-6XHis have been highlighted in two section of the discussion.
From Line 353
“An important limitation of the study is that, despite using codon optimisation in designing our constructs for expression in S. cerevisiae under control of the gain-of-function pdr1-3 transcriptional regulator in acting at the PDR5 locus, only modest expression of CnCYP51 was obtained. The recombinant strains were at least twice as sensitive to the triazole drugs FLC, VCZ, ITC and PCZ as the control yeast host strain (strain Y2411) which expressed the endogenous ScCYP51 (Erg11). The modest expression detected might be due to a problem caused by codon usage choice resulting in misfolding bottlenecks. In addition, evolutionary distance might affect enzyme turnover due to a sub-optimal membrane microenvironment or substrate preference i.e., a likely preference for eburicol over the lanosterol provided by the yeast expression system [57]. Despite a low level of CnCYP51 expression and the obligatory use of endogenous lanosterol as substrate, the normal growth rates, morphology and susceptibility to the control drug MCF of the recombinant cells and the inability to detect any suppressor strains on exposure to azole drugs suggests that the CnCYP51 constructs are not under genetic stress sufficient to modify susceptibility via increased azole target expression.
Another possible limitation is that the recombinant CnCPR is much more highly expressed than CnCYP51. However, overexpression of the cognate CnCPR as a CnCPR-6×His construct resulted in significant differences in the susceptibility of the co-expressing strains to azoles without increasing CnCYP51 expression levels.”
Additional minor corrections highlighted
Line 2 neoformans
Line 4 Both affiliation of M V Keniya given
Line 124 Grammar in sentence corrected
Line 173 Molecular weight of endogenous ScCYP51 given
Line 204 Commercial name of VT-1161 (Otesoconazole) given
Lines 245, 253 and 254 Volumes in elution profile corrected
Line 348 Sentence simplified to summarize conclusion of previous sentence
Table 2 Compound MCC8186 deleted as not mentioned in text

Round 2
Reviewer 1 Report
The manuscript has been corrected taking into account the reviewers' comments